# The lymphocyte-specific protein tyrosine kinase-specific inhibitor A-770041 attenuates lung fibrosis via the suppression of TGF-β production in regulatory T-cells

**Kozo Kagawa[1], Seidai Sato[1], Kazuya Koyama[1], Takeshi Imakura[1], Kojin Murakami[1], Yuya Yamashita[1], Nobuhito Naito[1], Hirohisa Ogawa[2], Hiroshi Kawano[1], Yasuhiko Nishioka**[1]*

**1** Department of Respiratory Medicine and Rheumatology, Graduate School of Biomedical Sciences, Tokushima University, Tokushima, Japan, **2** Department of Pathology and Laboratory Medicine, Graduate School of Biomedical Sciences, Tokushima University, Tokushima, Japan

* yasuhiko@tokushima-u.ac.jp

**Data Availability Statement:** All relevant data are within the paper and its Supporting Information files.

## Abstract

### Background

Lymphocyte-specific protein tyrosine kinase (Lck) is a member of the Src family of tyrosine kinases. The significance of Lck inhibition in lung fibrosis has not yet been fully elucidated, even though lung fibrosis is commonly preceded by inflammation caused by infiltration of T-cells expressing Lck. In this study, we examined the effect of Lck inhibition in an experimental mouse model of lung fibrosis. We also evaluated the effect of Lck inhibition on the expression of TGF-β1, an inhibitory cytokine regulating the immune function, in regulatory T-cells (Tregs).

### Methods

Lung fibrosis was induced in mice by intratracheal administration of bleomycin. A-770041, a Lck-specific inhibitor, was administrated daily by gavage. Tregs were isolated from the lung using a CD4+CD25+ Regulatory T-cell Isolation Kit. The expression of *Tgfb* on Tregs was examined by flow cytometry and quantitative polymerase chain reaction. The concentration of TGF-β in bronchoalveolar lavage fluid (BALF) and cell culture supernatant from Tregs was quantified by an enzyme-linked immunosorbent assay.

### Results

A-770041 inhibited the phosphorylation of Lck in murine lymphocytes to the same degree as nintedanib. A-770041 attenuated lung fibrosis in bleomycin-treated mice and reduced the concentration of TGF-β in BALF. A flow-cytometry analysis showed that A-770041 reduced the number of Tregs producing TGF-β1 in the lung. In isolated Tregs, Lck inhibition by A-770041 decreased the *Tgfb* mRNA level as well as the concentration of TGF-β in the supernatant.

**Funding:** This work was supported by a grant to the Ministry of Health, Labour and Welfare, the Study Group on Diffuse Pulmonary Disorders, Scientific Research/Research on Intractable Diseases (Y.N.; Code Number: 20FC1033) and a grant from Boehringer-Ingelheim. Boehringer-Ingelheim reviewed this manuscript. The funders had no role in study design, data collection and analysis, decision to publish, or preparation of the manuscript.

**Competing interests:** The authors have declared that no competing interests exist.

**Abbreviations:** IPF, idiopathic pulmonary fibrosis; ECM, extracellular matrix; PDGFR, platelet-derived growth factor receptor; VEGFR, vascular endothelial growth factor receptor; FGFR, fibroblast growth factor receptor; BLM, bleomycin; Lck, Lymphocyte-specific protein tyrosine kinase; IC$_{50}$, the half maximal inhibitory concentration; TCR, T-cell receptor; BAL, bronchoalveolar lavage; BALF, BAL fluid; Tregs, regulatory T-cells; qPCR, quantitative PCR; H&E stain, hematoxylin and eosin stain; Abs, antibodies; MACS, magnetic activated cell sorting; IFN−γ, interferon-γ.

## Conclusions

These results suggest that Lck inhibition attenuated lung fibrosis by suppressing TGF-β production in Tregs and support the role of Tregs in the pathogenesis of lung fibrosis.

## Introduction

Idiopathic pulmonary fibrosis (IPF) is a chronic disease characterized by progressive loss of lung tissue, with a median survival time of two to four years after the diagnosis [1, 2]. Characteristic pathological features of IPF are repetitive microscopic alveolar epithelial cell injury and dysregulated repair, fibrosis, and excessive deposition of extracellular matrix (ECM), resulting in the loss of the parenchymal architecture and lung function [3, 4].

Nintedanib has been approved for the treatment of IPF in several countries and regions [5]. A number of clinical trials have shown that treatment with nintedanib led to a reduction in the annual rate of decline of the forced vital capacity in patients with IPF [6–8]. Nintedanib is a potent intracellular inhibitor of the receptor tyrosine kinases targeting platelet-derived growth factor receptor (PDGFR), vascular endothelial growth factor receptor (VEGFR), fibroblast growth factor receptor (FGFR), and non-receptor tyrosine kinases of the Src family [9, 10]. These tyrosine kinases targeted by nintedanib have been reported to play a critical role in the pathogenesis of IPF [11]. For example, the increased release of PDGF and FGF are observed in the lung tissue of IPF patients [11], and several groups, including our own, have reported that blocking PDGFR signaling ameliorates lung fibrosis in experimental animal models [12–15]. Furthermore, the targeting of VEGFR signaling reportedly attenuates lung fibrosis [16, 17]. Recent data have also shown that, in bleomycin (BLM)-treated mice, nintedanib inhibits fibroblast proliferation [18] and differentiation and migration of fibrocytes [19] and prevents polarization of M2 macrophages [20]. However, while all of these signaling pathways are considered major mediators of lung fibrosis, analyses of their respective effects have focused on the inhibitory effects of nintedanib on FGFR, PDGFR, and VEGFR.

In addition to the above, nintedanib also has an inhibitory effect on lymphocyte-specific protein tyrosine kinase (Lck). The IC$_{50}$ value of nintedanib against Lck is reported to be 22 nM, which is lower than that against FGFR, PDGFR, and VEGFR [18].

Lck is a member of the Src family of tyrosine kinases that is predominantly expressed in T-cells [21, 22]. Lck plays a critical role in the early propagation and regulation of the T-cell development and homeostasis by modulating T-cell receptor (TCR) signaling [23]. The role of T-cells in fibrosis is poorly understood. However, relative to samples obtained from normal individuals, lung tissue and bronchoalveolar lavage (BAL) fluid from patients with IPF are reported to be enriched in several populations of T-cells [24]. Furthermore, blocking T-cell influx by anti-CD3 antibodies reportedly abrogated lung fibrosis in an experimental mouse model, suggesting that T-cells are a key participant in lung fibrosis [25]. As such, T-cells might also be involved in lung fibrosis via an as-yet-undefined mechanism and may be a therapeutic target of nintedanib through its Lck-inhibiting effect. However, the significance of Lck inhibition in lung fibrosis has not yet been fully elucidated.

Therefore, in this study, we focused on the expression of Lck by T-cells and examined the effect of Lck inhibition using the Lck-specific inhibitor A-770041 in a mouse model of experimental lung fibrosis. We also evaluated the effect of Lck inhibition on the expression of TGF-β1, an inhibitory cytokine that regulates the immune function, in regulatory T-cells (Tregs).

## Materials and methods

Detailed methods are described in the online supplement.

### Mice and agents

Eight-week-old male C57BL/6 mice were purchased from Charles River Japan Inc. (Tokyo, Japan). All experimental protocols were approved by the animal research committee of the University of Tokushima, Japan. A-770041 and nintedanib were kindly provided by Boehringer Ingelheim (Ingelheim, Germany).

### CD4$^+$ T-cells and Tregs isolation

CD4$^+$ T-cells and Tregs were obtained from murine lung or spleen. CD4$^+$ T-cells and Tregs were isolated using auto-MACS with CD4$^+$ T-cell Isolation Kits and with CD4$^+$ CD25$^+$ Regulatory T-cell Isolation Kits, respectively.

### Analyses of the phosphorylation of Lck

CD4$^+$ T-cells were incubated with beads coated with CD3/CD28 antibody and with RPMI with various concentrations of nintedanib or A-770041. After stimulation with CD3/CD28 antibody, CD4$^+$ T-cells were collected and lysed immediately, and the phosphorylation of Lck was determined using the Simple Western$^{TM}$ System (ProteinSimple, Santa Clara, CA, USA), according to a previous report [26].

### BLM-induced lung fibrosis in mice

Mice received a single transbronchial instillation of 3.0 mg/kg BLM on day 0 as previously described [19]. A-770041 (5 mg/kg), nintedanib (60 mg/kg), or distilled water was administered daily by gavage. The dosage of A-770041 was determined based on previous study [27]. In experiments to evaluate the effect of A-770041 on BLM treated murine lungs, mice were separately treated with A-770041 from days 0 to 10 (early phase), days 11 to 21 (late phase), or days 0 to 21 (full treatment).

### The hydroxyproline assay

The lungs were homogenized in distilled water, and the hydroxyproline contents were measured using a Bio-vision hydroxyproline assay kit (BioVision, Mount View, CA, USA).

### Histopathology

BLM-treated lungs were harvested, fixed in 10% formalin, and embedded in paraffin. Three-micrometer-thick sections were stained with hematoxylin and eosin (H&E) stain or azan Mallory. In the quantitative analysis, a numeric fibrotic scale was used (Ashcroft score) [28].

### BAL

BAL was performed for both lungs with saline (1 ml).

### Flow-cytometry analyses

The cells were stained with conjugated antibodies (Abs) and permeabilized with a Fixation/Permeabilization Kit according to the manufacturer's instructions (BD Biosciences, San Diego, CA, USA), before being stained with Abs to IL-17, TGF-β1, and Foxp3.

## Analyses of the expression and production of TGF-β in Tregs

Tregs were incubated with beads coated with CD3/CD28 antibody and with RPMI with various concentration of A-770041 (100, 500 nM) for 24 h. Supernatants were collected and used for an enzyme-linked immunosorbent assay (ELISA) to analyze the TGF-β concentration, as described below. To evaluate the mRNA expression of Tgfβ, cells were also lysed and used for reverse transcription polymerase chain reaction (RT-PCR).

## TGF-β concentrations

The concentrations of TGF-β in the BAL fluid (BALF) and supernatants of cell cultures were examined using an ELISA kit purchased from R&D Systems (Minneapolis, MN, USA).

## Quantitative PCR (qPCR)

qPCR was performed as previously described [29].

## Statistical analyses

The significance of differences was analyzed using a one-way analysis of variance, followed by Tukey's multiple-comparison post-hoc test. $p$ values of less than 0.05 were considered significant. Statistical analyses were performed using the GraphPad Prism software program Ver. 6.01 (GraphPad Software Inc., San Diego, CA, USA).

# Results

## A-770041 inhibits the phosphorylation of Lck in murine lymphocytes

First, we examined the effect of nintedanib and A-770041 on the phosphorylation of Lck in murine CD4+ T-cells. CD4+ T-cells were isolated from murine spleen using magnetic activated cell sorting (MACS). The phosphorylation of Lck of lymphocytes was inhibited by nintedanib mainly at a ≥30 nM (Fig 1A and 1B). Similarly, the inhibitory effect of A-770041 was observed at ≥100 nM (Fig 1C and 1D). These results suggest that A-770041 is as effective as nintedanib in inhibiting the phosphorylation of Lck.

## A-77041 attenuates BLM-induced lung fibrosis in mice

To examine how Lck inhibition works *in vivo*, we induced fibrosis in the mouse lung by BLM. After mice received a single transbronchial instillation of BLM on day 0, A-770041 was administrated daily by gavage. To examine the time kinetics of the antifibrotic effects of A-770041, mice were separately treated with A-770041 from days 0 to 10 (early phase), days 11 to 21 (late phase), or days 0 to 21 (full treatment). On day 21, the mice were killed, and fibrotic changes in the lungs were assessed using the Ashcroft scoring system and a hydroxyproline assay.

The number of fibrotic lesions in the lungs of BLM-treated mice was reduced in the early-phase and full-treatment groups but not in the late-phase group (Fig 2A). A quantitative histological analysis showed that the Ashcroft fibrotic score in the early-phase and full-treatment groups was significantly lower than in those treated with BLM alone (Fig 2B). A hydroxyproline colorimetric assay also showed that the collagen content was reduced in the lungs of the early-phase and full treatment groups but not in the late-phase group (Fig 2C).

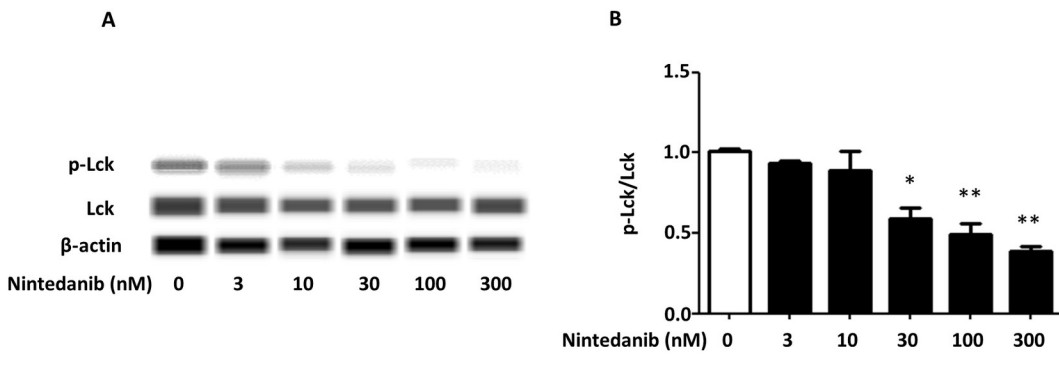

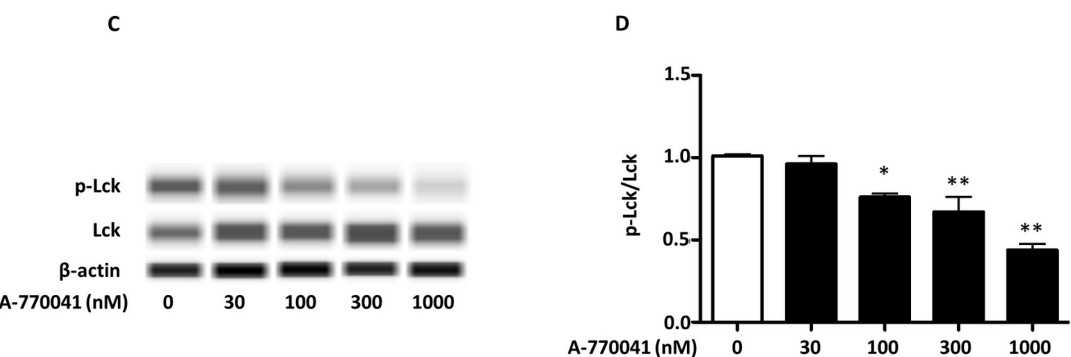

**Fig 1. Nintedanib and A-770041 inhibit the phosphorylation of lymphocyte-specific protein tyrosine kinase (Lck) of murine CD4+ T-cells.** CD4+ T-cells were collected from murine spleens using magnetic activated cell sorting. CD4+ T-cells were stimulated by anti-CD3/CD28 antibodies and incubated with different concentrations of nintedanib (0–300 nM) (A)(B) or A-770041 (0–1000 nM) (C)(D) for 5 minutes. The phosphorylation of Lck was analyzed by the Simple Wes™ system (n = 4). *p<0.05 versus groups treated without nintedanib or A-770041; **p<0.01 versus groups treated without nintedanib or A-770041.

## A-770041 reduces the infiltration of TGF-β-producing CD4+ T-cells into the lungs

Our next goal was to find a logical reason as to why Lck inhibition of lymphocytes was effective in a lung fibrosis model. Before evaluating the effect of A-770041 on lymphocytes *in vivo*, we assessed the dynamics of lymphocytes in a mouse model of BLM-induced lung fibrosis. After receiving a single transbronchial instillation of BLM, BALF was collected and analyzed on days 7, 14, and 21. The number of lymphocytes in BALF peaked on day 14 (Fig 3A). Flow-cytometry analyses revealed the CD4/8 ratio gradually increased after BLM instillation (Fig 3C).

Lymphocytes have been reported to contribute to the progression of lung fibrosis by secreting profibrotic cytokines. Well-known lymphocyte-derived cytokines that enhance tissue fibrosis include interferon-γ (IFN-γ), IL-17, and TGF-β [30]. Among them, the role of IFN-γ is controversial, as although IFN-γ gene-deleted mice show a reduction in lung fibrosis [31, 32], the administration of IFN-γ-neutralizing antibodies is reported to enhance fibrotic changes in murine models [33]. Therefore, we analyzed the proportion of IL-17-producing CD4+ T-cells and TGF-β-producing CD4+ T-cells in a single-cell suspension derived from lung tissue of a mouse model of BLM-induced lung fibrosis.

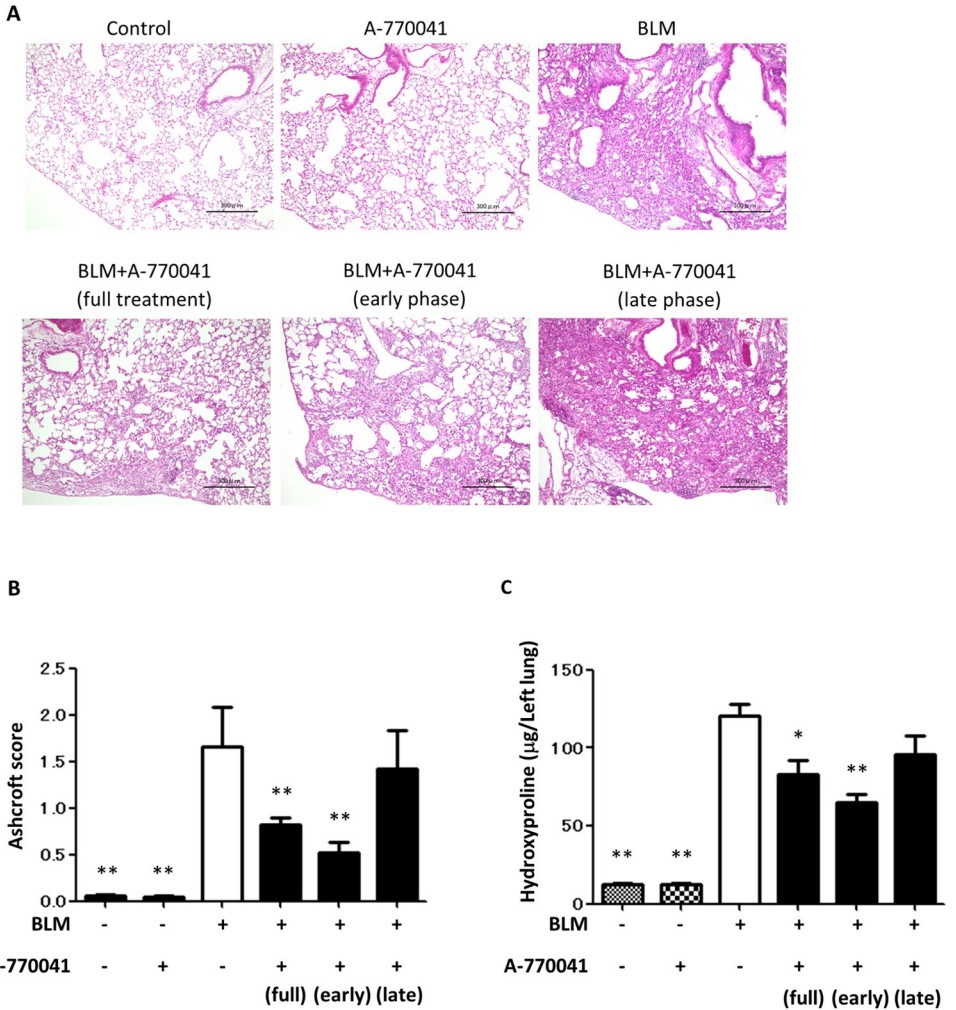

**Fig 2. A-770041 attenuates BLM-induced lung fibrosis.** Mice received a single transbronchial instillation of 3.0 mg/kg BLM on Day 0, and A-770041 (5 mg/kg/d) or vehicle was administered daily. Mice were separately treated with A-770041 from days 0 to 10 (early phase), days 11 to 21 (late phase) or days 0 to 21 (full treatment). On Day 21, mice were killed, and their lungs were collected. Lung sections were stained with (A) hematoxylin and eosin. (B) The fibrotic changes in the lungs were quantified with a numerical fibrotic score (Ashcroft score) histopathologically (n = 6). (C) The hydroxyproline contents in lung tissue were measured by a hydroxyproline assay (n = 6). *p<0.05 versus groups treated with BLM without A-770041; **p<0.01 versus groups treated with BLM without A-770041.

Flow-cytometry analyses revealed that the proportion of TGF-β-producing CD4+ T-cells increased from days 7 to 21 after BLM instillation (Fig 3B and 3E). The proportion of IL-17-producing CD4+ T-cells was slightly increased on day 14 after BLM instillation, but not to a significant degree (Fig 3D).

We next examined whether or not Lck inhibition affected these two CD4+ T-cell populations. IL-17- and TGF-β-producing CD4+ T-cells were isolated from lung tissue of BLM-treated mice. Despite the mice having been administered A-770041, the percentage of IL-17-producing CD4+ T-cells was not markedly changed (Fig 3F). In contrast, TGF-β-producing CD4+ T-cells in the lung were significantly decreased by A-770041 (Fig 3G). The concentration of TGF-β in the BALF of BLM-treated mice was also decreased by the administration of A-770041 (Fig 3H). These results suggest that TGF-β-producing CD4+ T-cells are the therapeutic targets of Lck inhibition by A-770041.

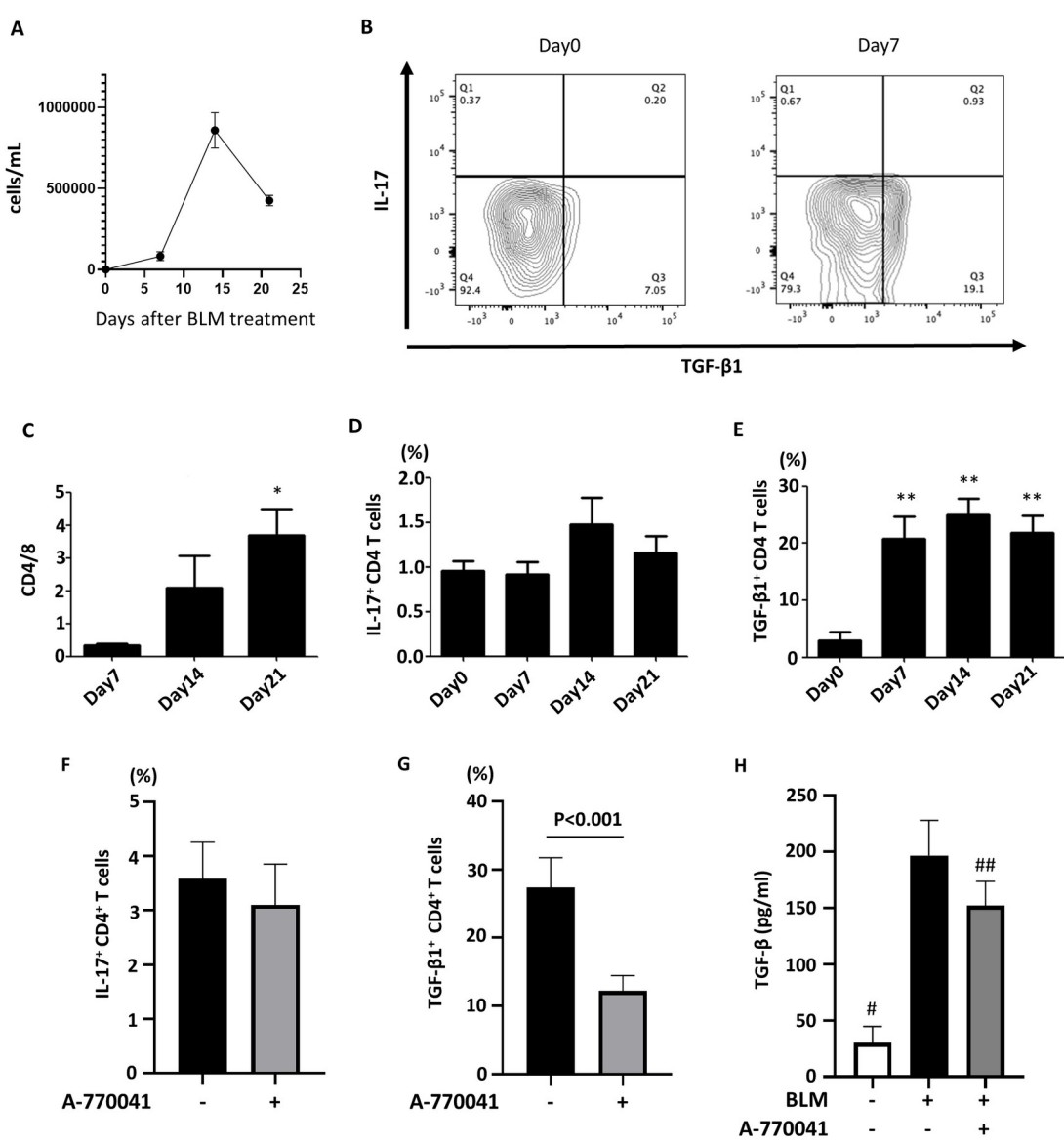

A770041 compared to the non-treated group *p<0.05, **<0.01

**Fig 3. A-770041 reduces the percentage of TGF-β-producing CD4+ T-cells in BLM treated lungs.** Mice received a single transbronchial instillation of 3.0 mg/kg BLM on Day 0, and BALF was collected on days 0, 7, 14, and 21. Lymphocyte counts (A) and the CD4/8 ratio (C) in the BALF were examined by flow cytometry (n = 5). The percentages of Il-17$^+$ CD4$^+$ T-cells and TGF-β1$^+$ T-cells infiltrating lung tissues were examined by flow cytometry (B, D, E). After BLM treatment, A-770041 (5 mg/kg/day) or vehicle was administered daily, and lungs were collected on day 7 (n = 5). The percentages of Il-17+ CD4+ T-cells and TGF-β1$^+$ CD4$^+$ T-cells infiltrating lung tissues with or without A-770041 were examined by flow cytometry (F, G). The concentration of TGF-β in BALF was examined by an ELISA. $^*$: p<0.05 versus samples of day 7; $^{**}$: p<0.01 versus samples of day 0; #: p<0.01 versus group treated with BLM without A-770041; ##: p<0.05 versus group treated with BLM without A-770041.

## A-770041 inhibited the production of TGF-β in Tregs

TGF-β is required for the maintenance and differentiation of Foxp3$^+$ Tregs peripherally [34] and is produced by Tregs themselves in an autocrine or paracrine manner [35]. Therefore, we next examined the effect of Lck inhibition on Tregs in a BLM-induced lung fibrosis murine model.

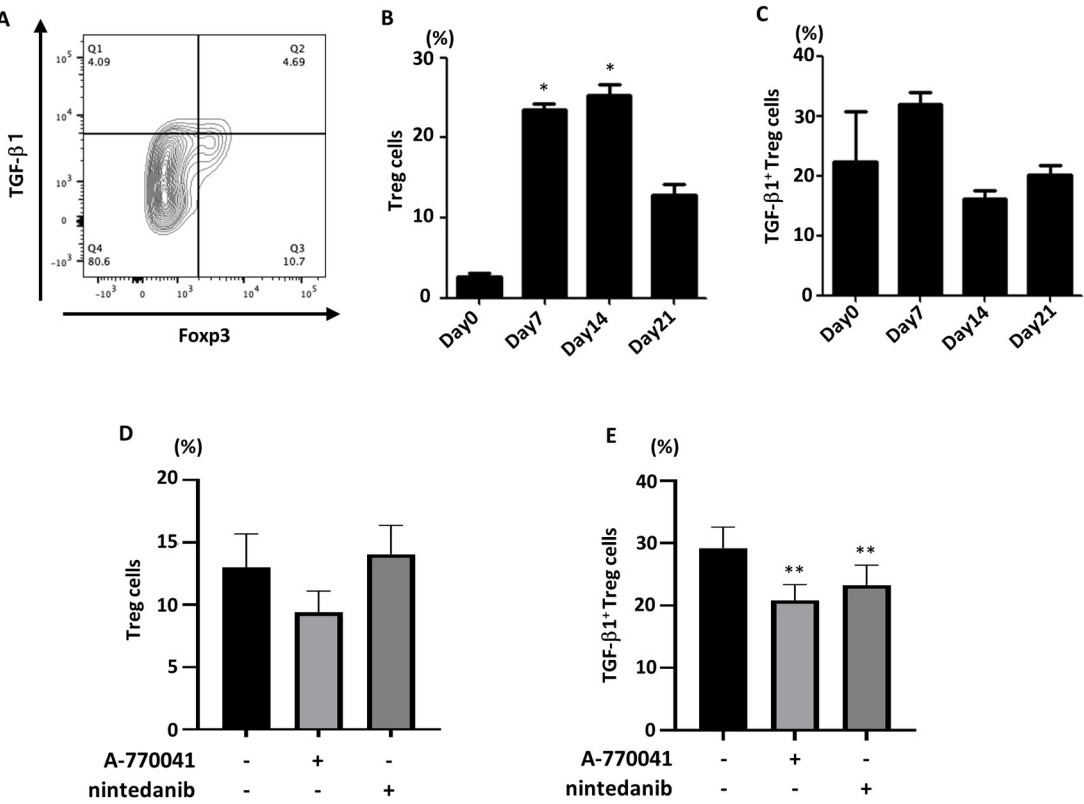

**Fig 4. A-770041 and nintedanib inhibit the production of TGF-β in regulatory T-cells.** Tregs and TGF-β1-producing Tregs infiltrating BLM-treated lung tissues were examined by flow cytometry. Tregs were identified as CD3e⁺ CD4⁺ Foxp3⁺ TGF-β1⁺ cells, and the gating strategy for the identification of TGF-β1-producing Tregs is shown (A). Mice received a single transbronchial instillation of 3.0 mg/kg BLM on Day 0, and lungs were collected on days 0, 7, 14, and 21. Time courses of the percentages of whole Treg cells to CD4+ T-cells (B) and that of TGF-β1+ Tregs to Tregs (C) were evaluated (Day 0, n = 4; Day 7, n = 8; Day 14, n = 8; Day 21, n = 6). After BLM treatment, A-770041 (5 mg/kg/day), nintedanib (60 mg/kg/day), or vehicle was administered daily, and lungs were collected on day 7 (n = 5). The percentages of whole Tregs to CD4⁺ T-cells (D) and that of TGF-β1⁺ Tregs to Tregs (E) were evaluated by flow cytometry. *p<0.05 versus Day 0 samples; **p<0.05 versus group treated with BLM without A-770041 or nintedanib.

Before evaluating the effect of A-770041 on Tregs *in vivo*, we assessed the time-course of the appearance of Tregs in the lung tissue of a mouse model of BLM-induced lung fibrosis. Similar to TGF-β-producing CD4⁺ T-cells, Treg cells were also increased in mouse lung starting at seven days after BLM instillation, although the ratio of the TGF-β-positive population among these cells was not markedly changed (Fig 4A–4C). The administration of A-770041 did not affect the number of Tregs in the lung (Fig 4D). However, the percentage of TGF-β-producing Tregs was significantly decreased by A-770041 (Fig 4E). The same effect was observed with the administration of nintedanib (Fig 4D and 4E). Taken together, these results suggest that A-770041 reduces the amount of TGF-β in the lung by suppressing the expression of TGF-β in Tregs.

We next examined the effect of Lck inhibition on Tregs *in vitro*. Tregs were isolated from mouse spleen and incubated with A-770041. A-770041 decreased the *Tgfb* mRNA level in isolated Tregs and the concentration of TGF-β in the culture supernatants of isolated Tregs (Fig 5A and 5B).

## Discussion

In this study, we examined the role of Lck inhibition on lung fibrosis using an experimental mouse model. The results from this study suggest that early-phase treatment with A-770041, a

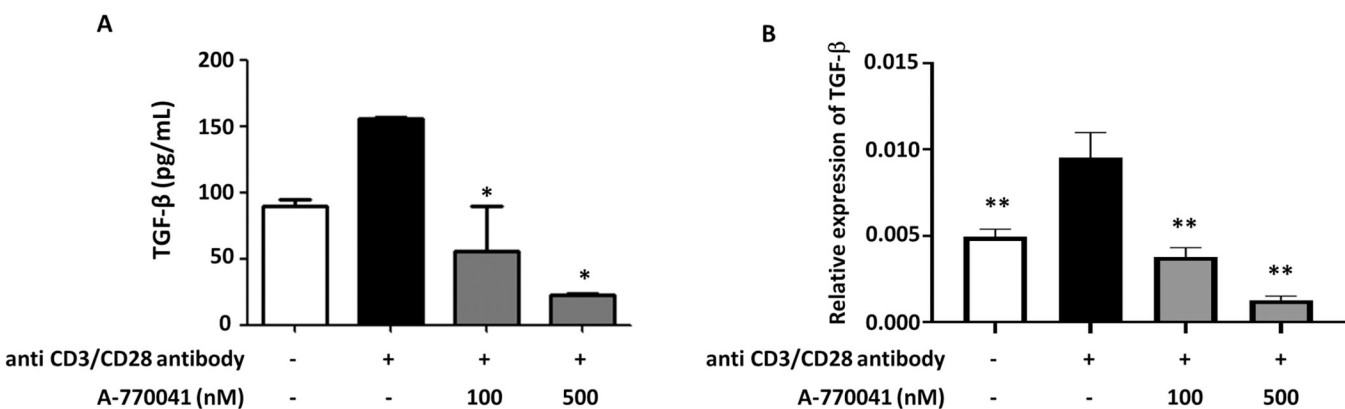

**Fig 5. A-770041 inhibit the production of TGF-β in Tregs *in vitro*.** Tregs obtained from murine spleen were stimulated by anti-CD3/CD28 antibodies and incubated with different concentrations of A-770041 (0, 100, 500 nM) for 24 h. Concentrations of TGF-β in the supernatant were determined by an ELISA (n = 4). The expression of *Tgfb* mRNA in incubated cells was examined by qPCR (n = 4).

Lck-specific inhibitor, significantly mitigated lung fibrosis induced by BLM. In addition, we also showed that A-770041 decreased the concentration of TGF-β in BALF collected from a BLM-induced lung fibrosis model via the inhibition of TGF-β production by Tregs.

Previous studies had failed to clarify the significance of Lck inhibition in the pathology of lung fibrosis. Nintedanib, a small-molecule tyrosine kinase inhibitor with Lck-inhibiting activity, has anti-inflammatory properties and has been shown to reduce lymphocytes in BALF from two types of animal models of experimental lung fibrosis [18]. In addition, a study using blood cells isolated from the peripheral blood of healthy donors showed that nintedanib blocked T-cell activation by inhibiting Lck-Y394 phosphorylation [36]. These previous reports suggest that the therapeutic effect of nintedanib on lung fibrosis, including its anti-inflammatory effects, may be due in part to its inhibitory activity on Lck. However, because nintedanib, which is a multi-targeted tyrosine kinase inhibitor, also has inhibitory effects on tyrosine kinases involved in inflammation other than Lck, such as Src [18, 37], the present study is the first report to examine the effect of Lck-specific inhibition on lung fibrosis.

Our study focused on the effect of Lck inhibition on T-cell subsets and found that Lck inhibition by A-770041 reduced the ratio of TGF-β⁺ CD4⁺ cells via a reduction in the TGF-β concentration in BALF. Interestingly, both A-770041 and nintedanib reduced TGF-β production in Tregs, which are considered to be the major TGF-β-producing lymphocytes, but did not decrease the number of Tregs. Similarly, Ubieta et al. also reported that nintedanib did not affect the percentages or cell numbers of various T-cell subsets, including CD4⁺, CD8⁺, Th1, Th2, and Tregs [36].

In contrast, Tanaka et al. reported that imatinib, which also has an inhibitory effect on Lck, selectively causes apoptosis in Tregs [38]. Furthermore, they also reported that AMG-47a, a Lck inhibitor, selectively reduced the number of effector Tregs [38]. However, although the $IC_{50}$ values for Lck of imatinib and AMG-47a were reported to be 0.3 mM and 0.2 nM, respectively [39, 40], both compounds were used at therapeutic doses for chronic myelogenous leukemia to Tregs in their study. The half-maximal effective concentrations of imatinib and AMG-47a for apoptosis of effector Tregs were reported to be 8.8 mM and 411.8 nM, respectively [38]. Furthermore, AMG-47a inhibits not only Lck but also VEGF2, p38α, Jak3, MLR, and IL-2, with $IC_{50}$ values of 1 nM, 3 nM, 72 nM, 30 nM, and 21 nM, respectively [40]. Therefore, the different doses and compounds used for Lck inhibition might have been responsible for the different effects on Tregs between the present and previous study.

In the present study, Lck inhibition suppressed TGF-β expression in CD4+ T-cells and Tregs at both mRNA and protein levels. Lck is crucial for the initiation of downstream of TCR signaling [41] and consequently regulates the activation of Erk1/2 and Akt, which are downstream factors in TCR signaling [42]. On the other hand, it has been reported that Erk is a crucial factor in TGF-β transcription and that Akt activation is required for TGF-β translation [43, 44]. Therefore, Lck inhibition may have suppressed TGF-β transcription and translation via inhibition of ERK and Akt activation. Furthermore, Lck binds to and phosphorylates FoxP3, the master transcriptional factor of Tregs, enhancing its inhibitory activity. FoxP3 has been reported to suppress the expression of SMAD7 which exerts negative feedback on TGF-β signaling [45]. TGF-β signaling positively regulate TGF-β gene expression by autocrine [44]. Thus, Lck inhibition may suppress TGF-β signaling and TGF-β gene expression via upregulation of SMAD7 expression following suppression of FoxP3, especially in Tregs. However, the mechanism of how Lck inhibition affects TGFβ expression needs further investigation.

The role and function of Tregs in lung fibrosis is not fully understood. A previous report showed a decrease in the number of CD4$^+$CD25$^+$FoxP3$^+$Tregs in the lungs and blood of IPF patients [46], while subsequent studies have shown the opposite [47, 48]. In addition to a previous report on Tregs impairment in IPF [46], there is a report that the immunosuppressive function of Tregs is important in suppressing lung fibrosis, as the administration of dysfunctional Tregs worsened fibrosis in a TGF-β1-induced mouse model of lung fibrosis compared to the administration of normal Tregs [48]. In contrast, in a silica-induced murine model of lung fibrosis, the neutralization of immunosuppressive activity of Tregs reportedly led to the accumulation of effector T-cells and contributed to the worsening of fibrosis via IL-4 secretion [49]. In animal models, one report showed that lung injury may induce Tregs alteration, which can augment lung fibrosis [50], while another reported a protective role against fibrosis [51]. Based on these results, it is currently suggested that Tregs may demonstrate functional heterogeneity in response to the lung microenvironment and adopt differential roles in lung fibrosis that depend on the combinatorial influence of intrinsic cellular properties and their response to the local milieu [52].

Interestingly, it has been suggested that Tregs play a detrimental role in the early stages but a protective role in the late stages of lung fibrosis in mice [53]. Therefore, given the present finding that the early-phase administration of A-770041, but not the late-phase, ameliorated BLM-induced lung fibrosis, the therapeutic effect of Lck inhibition in Tregs in the early stage of lung fibrosis may also be due in part to changes in the Treg function. However, because TGF-β is a potent chemical mediator of lung fibrosis, with prior studies showing that Tregs promote fibrosis via TGF-β-associated mechanisms, the effect of Lck inhibition in the present study to reduce TGF-β production by Tregs [49] may be advantageous for lung fibrosis, regardless of the timing of treatment. Why late-phase Lck inhibition did not sufficiently suppress lung fibrosis may be because the suppression of TGF-β production in Tregs alone was not sufficient, as TGF-β production by non-lymphocytes, such as epithelial cells, is increased in the fibrosis phase [54].

The present study was associated with some limitations. First, the effect of Lck inhibition on cell types other than T-cells has not been fully investigated. Because Lck expression is generally specific to lymphocytes, the effect on other cell types is expected to be small. However, we cannot rule out the possibility that the effect of Lck inhibition on other types of T-cells, such as helper T-cells and effector T-cells, might have affected the results. Second, only one experimental mouse model was used in this study. Third, the specificity of A-770041 as a Lck inhibitor may have been a limitation, as A-770041 has been reported to have not only strong selective activity for Lck but also very weak selective activity for other Src family tyrosine

kinases, including Src, Fyn, Fgr, HCK, and Tie2 [27, 55] Therefore, these other inhibitory effects might have affected the results in the present study.

In summary, Lck inhibition attenuated lung fibrosis by suppressing TGF-β production in Tregs. It was suggested that part of the anti-fibrotic effect of nintedanib may also have been due to Lck inhibition. These results support the role of lymphocytes in the early stage of lung fibrosis and suggest that A-770041 might be useful for ameliorating lung inflammation induced by fibrotic injury and subsequent fibrogenesis.

## Supporting information

**S1 File. The detailed method.**
(DOCX)

**S1 Fig. Raw data of representative analysis result shown in Fig 1A.** Phosphorylated Lck (a), Lck (b) and β-actin (c) were determined by a Simple WesternTM System. The vertical axis of each graph shows the fluorescence intensity of each molecular weight protein.
(PDF)

**S2 Fig. Raw data of representative analysis result shown in Fig 1C.** Phosphorylated Lck (a), Lck (b) and β-actin (c) were determined by a Simple WesternTM System. The vertical axis of each graph shows the fluorescence intensity of each molecular weight protein.
(PDF)

## Acknowledgments

The authors thank Ms. Akie Tanabe for her technical assistance. We also thank the members of the Nishioka lab for their technical advice and fruitful discussions. This study was supported by Support Center for Advanced Medical Sciences, Tokushima University Graduate School of Biomedical Sciences.

## Author Contributions

**Conceptualization:** Kozo Kagawa, Seidai Sato, Kazuya Koyama, Yasuhiko Nishioka.

**Data curation:** Kozo Kagawa, Kazuya Koyama.

**Funding acquisition:** Yasuhiko Nishioka.

**Investigation:** Kozo Kagawa, Seidai Sato, Kazuya Koyama, Takeshi Imakura, Kojin Mura-kami, Yuya Yamashita, Nobuhito Naito, Hirohisa Ogawa, Hiroshi Kawano, Yasuhiko Nishioka.

**Project administration:** Seidai Sato, Yasuhiko Nishioka.

**Supervision:** Seidai Sato, Yasuhiko Nishioka.

**Visualization:** Kozo Kagawa, Kazuya Koyama.

**Writing – original draft:** Seidai Sato.

**Writing – review & editing:** Yasuhiko Nishioka.

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
