## [Decision Letter · Decision Letter 0]

30 Aug 2022

PONE-D-22-18289The lymphocyte-specific protein tyrosine kinase-specific inhibitor A-770041 attenuates lung fibrosis via the suppression of TGF-β production in regulatory T-cellsPLOS ONE

Dear Dr. Nishioka,

Thank you for submitting your manuscript to PLOS ONE. After careful consideration, we feel that it has merit but does not fully meet PLOS ONE’s publication criteria as it currently stands. Therefore, we invite you to submit a revised version of the manuscript that addresses the points raised during the review process.

Your manuscript was reviewed by a knowledgeable referee in the area. As noted in the attached comments, the reviewer has felt that the manuscript is technically sound, and the data support the conclusions. In addition, the manuscript is presented in an intelligible fashion and written in standard English. Although the reviewer has suggested acceptation of the manuscript, he/she recommends that the manuscript might benfit from additional insights with respect to the potential molecular mechanism how Lck inhibition results in reduced number of the TGFb-expressing CD4+ T cells and TGFb-expressing Tregs and the Tgfb mRNA level in isolated Treg. The timing of the different drug treatments (early, late, full) might be included in the methods section.

We look forward to receiving your revised manuscript.

Kind regards,

Laszlo Buday

Academic Editor

PLOS ONE

Journal Requirements:

"This work was supported by a grant to the Ministry of Health, Labour and Welfare, the Study Group on Diffuse Pulmonary Disorders, Scientific Research/Research on Intractable Diseases (Y.N.; Code Number: 20FC1033) and a grant from Boehringer-Ingelheim. Boehringer-Ingelheim reviewed this manuscript."

Reviewers' comments:

Reviewer's Responses to Questions

**Comments to the Author**

1. Is the manuscript technically sound, and do the data support the conclusions?

Reviewer #1: Yes

2. Has the statistical analysis been performed appropriately and rigorously? 

Reviewer #1: Yes

3. Have the authors made all data underlying the findings in their manuscript fully available?

Reviewer #1: Yes

4. Is the manuscript presented in an intelligible fashion and written in standard English?

Reviewer #1: Yes

5. Review Comments to the Author

Reviewer #1: In this study Kagawa and colleagues elegantly explored the role of Lck inhibition by A-770041 on lung fibrosis using the experimental mouse model of bleomycin-induced lung inflammation and fibrosis. A-770041 was shown to inhibit the phosphorylation of Lck in murine spleen lymphocytes similar to nintedanib. Inhibition of Lck during the early inflammatory phase reduced lung fibrosis demonstrated by attenuated Ashcroft score and hydroxyproline content. In addition Lck inhibition decreased the concentration of TGF-β in BALF. A-770041 was shown to reduce the infiltration of TGF-β-producing CD4+ T-cells and Tregs into the lungs and to inhibit the production of TGF-β in Tregs. The publication further supports the hypothesis that part of the anti-fibrotic effect of nintedanib may also have been due to Lck inhibition.

The manuscript might benfit from additional insights with respect to the potential molecular mechanism how Lck inhibition results in reduced number of the TGFb-expressing CD4+ T cells and TGFb-expressing Tregs and the Tgfb mRNA level in isolated Treg.

The timing of the different drug treatments (early, late, full) might be included in the methods section.

6. PLOS authors have the option to publish the peer review history of their article (what does this mean?). If published, this will include your full peer review and any attached files.

Reviewer #1: No

---

## [Author Response · Author response to Decision Letter 0]

26 Sep 2022

Manuscript: # PONE-D-22-18289

Journal Requirements:

R1)

Thank you for your help. 

In the revised manuscript, File names, Corresponding Authorship, Heading, Figure Citations, Reference Citations, and Supporting Information Captions have been modified to comply with PLOS ONE style requirements.

We also changed the order of authorship (2nd author and 3rd author) with the consent of all authors.

"This work was supported by a grant to the Ministry of Health, Labour and Welfare, the Study Group on Diffuse Pulmonary Disorders, Scientific Research/Research on Intractable Diseases (Y.N.; Code Number: 20FC1033) and a grant from Boehringer-Ingelheim. Boehringer-Ingelheim reviewed this manuscript."

R2) 

Thank you for pointing out the issue regarding Funding.

The revised funding information is provided below.

Thank you for making the changes to the online submission form on our behalf.

Funding

This work was supported by a grant to the Ministry of Health, Labour and Welfare, the Study Group on Diffuse Pulmonary Disorders, Scientific Research/Research on Intractable Diseases (Y.N.; Code Number: 20FC1033) and a grant from Boehringer-Ingelheim. Boehringer-Ingelheim reviewed this manuscript. The funders had no role in study design, data collection and analysis, decision to publish, or preparation of the manuscript.

All the funding or sources of support (whether external or internal to your organization) received during this study are listed. There was no additional external funding received for this study.

R3)

In the revised version, we have added the raw data of Fig 1A and Fig 1C determined by a Simple WesternTM System in Supporting information as S1 Fig and S2 Fig. 

R4)

In the revised version, we have added the captions for Supporting Information at the end of manuscript.

R5) 

Some of the articles shown in the reference list were missing publication numbers and page numbers, which were reconfirmed and added. After that, we have verified that the reference list is complete and accurate.

Reviewers' comments:

Reviewer's Responses to Questions

Comments to the Author

Reviewer #1: In this study Kagawa and colleagues elegantly explored the role of Lck inhibition by A-770041 on lung fibrosis using the experimental mouse model of bleomycin-induced lung inflammation and fibrosis. A-770041 was shown to inhibit the phosphorylation of Lck in murine spleen lymphocytes similar to nintedanib. Inhibition of Lck during the early inflammatory phase reduced lung fibrosis demonstrated by attenuated Ashcroft score and hydroxyproline content. In addition Lck inhibition decreased the concentration of TGF-β in BALF. A-770041 was shown to reduce the infiltration of TGF-β-producing CD4+ T-cells and Tregs into the lungs and to inhibit the production of TGF-β in Tregs. The publication further supports the hypothesis that part of the anti-fibrotic effect of nintedanib may also have been due to Lck inhibition.

The manuscript might benfit from additional insights with respect to the potential molecular mechanism how Lck inhibition results in reduced number of the TGFb-expressing CD4+ T cells and TGFb-expressing Tregs and the Tgfb mRNA level in isolated Treg.

The timing of the different drug treatments (early, late, full) might be included in the methods section.

R) We strongly express our appreciate for the reviewer's comment on the potential molecular mechanism of how Lck inhibition affects TGF-β expression

In the present study, Lck inhibition suppressed TGF-β expression in CD4+ T-cells and Tregs at both mRNA and protein levels. Lck is crucial for the initiation of downstream of TCR signaling [Int J Mol Sci. 2019; 20:3500] and consequently regulates the activation of Erk1/2 and Akt, which are downstream factors in TCR signaling [Mol. Cancer Res. 2013;11:541–554]. On the other hand, it has been reported that Erk is a crucial factor in TGF-β transcription and that Akt activation is required for TGF-β translation [J Immunol. 2008;181: 3575–3585] [ PLoS One. 2011;6: e21465]. Therefore, Lck inhibition may have suppressed TGF-β transcription and translation via inhibition of ERK and Akt activation. 

Furthermore, Lck binds to and phosphorylates FoxP3, the master transcriptional factor of Tregs, enhancing its inhibitory activity. FoxP3 has been reported to suppress the expression of SMAD7 which exerts negative feedback on TGF-β signaling [J Immunol, 2004,172, 5149-5153]. TGF-β signaling positively regulate TGF-β gene expression by autocrine [PLoS One. 2011;6: e21465]. Thus, Lck inhibition may suppress TGF-β signaling and TGF-β gene expression via upregulation of SMAD7 expression following suppression of FoxP3, especially in Tregs. 

 We have added these hypotheses regarding potential molecular mechanism of how Lck inhibition affects TGF-β expression to the discussion section (page 14 line 6-19). 

 We also have added the timing of the different drug treatments (early, late, full) to the material and method section (page7 line 2-4).

---

## [Editor Report · Decision Letter 1]

27 Sep 2022

The lymphocyte-specific protein tyrosine kinase-specific inhibitor A-770041 attenuates lung fibrosis via the suppression of TGF-β production in regulatory T-cells

PONE-D-22-18289R1

Dear Dr. Nishioka,

We’re pleased to inform you that your manuscript has been judged scientifically suitable for publication and will be formally accepted for publication once it meets all outstanding technical requirements.

Kind regards,

Laszlo Buday

Academic Editor

PLOS ONE
---

## [Editor Report · Acceptance letter]

19 Oct 2022

PONE-D-22-18289R1 

The lymphocyte-specific protein tyrosine kinase-specific inhibitor A-770041 attenuates lung fibrosis via the suppression of TGF-β production in regulatory T-cells 

Dear Dr. Nishioka:

I'm pleased to inform you that your manuscript has been deemed suitable for publication in PLOS ONE. Congratulations! Your manuscript is now with our production department. 

Kind regards, 

on behalf of

Professor Laszlo Buday 

Academic Editor

PLOS ONE